# Waste Plastic Recycling Upgrade Design Nanogenerator for Catalytic Degradation of Pollutants

Qian Zhang [1,2], Qiyu Peng [2], Wenbin Li [2], Yanzhang Liu [2] and Xiaoxiong Wang [3,*]

1    National Engineering Laboratory for High-Speed Train System Integration, CRRC Qingdao Sifang Co., Ltd., Qingdao 266111, China; figozq100@sina.com
2    College of Mechanical and Electrical Engineering, Qingdao University, Qingdao 266071, China; sdrzpqy@163.com (Q.P.); liwenbin2022@163.com (W.L.); liuyanzhang0324@163.com (Y.L.)
3    College of Physics, Qingdao University, Qingdao 266071, China
*     Correspondence: wangxiaoxiong69@163.com

**Abstract:** In recent years, electrocatalytic degradation of pollutants based on nanogenerators has gradually emerged. Compared with the huge energy consumption of traditional electrocatalysis, this method can effectively use displacement current to induce charge transfer and complete catalysis, so it can directly use the existing water flow energy and other energy sources in nature. This work will explain the basic principles, methods, and measurement methods of preparing nanogenerators from waste plastics, as well as the classification of electrocatalytic principles and methods relative to nanogenerators, which provides important support for the research in this emerging field. At the same time, the analysis based on this knowledge will also lay the foundation for future design.

**Keywords:** nanogenerator; waste plastic; catalysis; electrocatalytic methods; catalytic degradation





## 1. Introduction

A nanogenerator converts mechanical energy into electrical energy. It attracts wide attention for its simple preparation process, which can be effectively achieved by using various forms of plastic, including recycled plastic products. In recent years, electrocatalytic degradation of pollutants based on nanogenerators has gradually emerged. Compared with the huge energy consumption of traditional electrocatalysis, this method can effectively use displacement current to induce charge transfer and complete catalysis, so it can directly use the existing water flow energy and other energy sources in nature. On the whole, the work of electrocatalytic degradation of pollutants based on recycled plastics is currently relatively scattered. Therefore, this work will explain the basic principles, methods, and measurement methods of preparing nanogenerators from waste plastics, as well as the classification of electrocatalytic principles and methods relative to nanogenerators, which provides important support for the research in this emerging field. At the same time, the analysis based on this knowledge will also lay the foundation for future design.

Plastic is a compound that is polymerized by addition polymerization or polycondensation reaction with monomer as the raw material. Over the past 100 years since its birth, this material has rapidly become widely used in all aspects of life [1–3]. However, the degradation problem caused by the excellent stability of plastics has become one of the most important environmental problems [4–6]. Especially in recent years, the penetration and pollution of microplastics in all aspects of human life have made the treatment of plastic pollution an urgent scientific problem [7,8]. At present, there are two ways to deal with the pollution of waste plastics. The first is to degrade plastics, including the use of degradable plastics, biological degradation, and chemical degradation [9–12]. The second is to recycle plastics and develop secondary products by using the widespread properties of plastics [13–15]. Figure 1 shows the variety of plastic pollutants in the ocean.

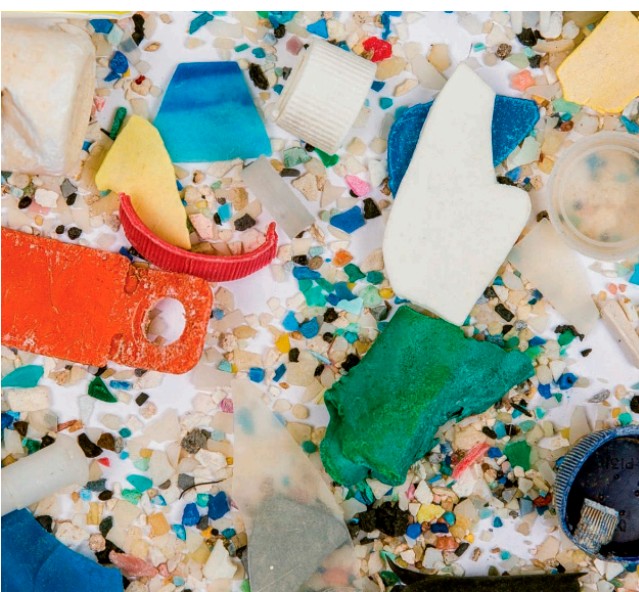

**Figure 1.** Plastic pollutants in the ocean. Reproduced from [16] with permission from the American Association Advancement of Science.

Nanogenerator (NG) is such a new and attractive research field. Generally, there are three types of NGs, namely piezoelectric nanogenerators (PENG), triboelectric nanogenerators (TENG), and pyroelectric nanogenerators (PyNG) [17–21]. PENG works on piezoelectric phenomena, and sufficient pressure on the material will induce current on the outer circuit with a relatively high voltage. To design this type of NG, piezoelectric materials will be required, such as polyvinylidene fluoride (PVDF) [22]. In this case, the materials should be polarized first, and a special PVDF was designed for better performance. Despite the struggle, PENGs have lower output and complex material design requirements. Despite its success in piezocatalysis, this type of NG still requires improvement for better application in recycled plastics. From this perspective, TENG has a wider material source and a more universal application environment. Meanwhile, such a material is reliable and has good performance, usually up to several tens of watts per square meter. The basic working principle of TENG is to carry out charge transfer through two materials with different electronegativity and then induce the external circuit to generate charge transfer through the relative position change of the material to convert mechanical energy into electrical energy. For TENG, the electronegativity of the material is the key to the good output of the design device. For plastic products, they can stably generate static electricity in a variety of friction environments, and using this static electricity to obtain power output can be very easily achieved. The most common example is the electrostatic discharge phenomenon when wearing a nylon cloth, indicating the wide existence of such a phenomenon. The last form of NG is PyNG; such NG converts heat into electric signals, requiring a cycled temperature change. Despite its rising tendency in the catalysis area, the recycled plastic design should be considered prior to the applications.

Electrocatalytic degradation of pollutants is one of the important methods in sewage treatment [23]. Through electrochemical design, environmentally harmful pollutants are degraded into environmentally harmless chemical compositions, including toxic substance degradation and colored dye degradation. However, traditional electrocatalytic degradation requires a lot of energy, and the ability to deal with actual environmental pollution problems is limited. Therefore, the design of new energy sources has become an important breakthrough direction in the field of development. Based on TENG, the direction of mechanical energy collection can be designed, including water flow energy collection, wind energy collection, etc., which can effectively use the existing energy in nature to convert it into electrical energy to degrade pollutants, thereby improving the relevant design effect.

## 2. The Basic Principles of Nanogenerators from Waste Plastics

Here, the working principles of the NGs are given individually. For scientific research, the three types of NGs can just be fabricated, and the NGs can be linked with electrolysis electrodes. In such a case, all NGs work as power sources. However, in real-world applications, the working principle should be studied before designing high-performance applications. Here, PENGs and TENGs collect mechanical energy, while PyNGs collect thermal energy. So, the most common scenarios of the former two are on the human body surface (movement), in wind, or in water (water flow or vibration), while PyNGs are applicable on the body surface or in flue holes.

### 2.1. PENG

PENG works depending on the piezoelectric effect. PENG's working mechanism is shown in Figure 2a. This effect is well known [24,25], and if the piezoelectric material is pressed, charge will emerge on both sides of the material. Basically, such charge emerged due to the induced charge screen, and if an electrode was fabricated, such charge movement would be turned into electric current in the case of an existing closed outer circuit. Here, lead zirconate titanate piezoelectric (PZT) and PVDF are representatives for inorganic and organic piezoelectric materials individually. PVDF is widely used in daily life, but such a material usually requires difficult polarization under high electric field cooling. PZT emerges in daily life; it can generate thousands of volts after easy polarization simply by an electric field [26]. The key factor for evaluating the performance of a piezoelectric material is its piezoelectric coefficient, $d_{33}$. The performance of a piezoelectric material can also be evaluated by measuring the charge generated by periodic pressing. Meanwhile, the polarization can also be measured by hysteresis loop measurement by dividing charge by area. Usually, materials with larger $d_{33}$ or polarization outputs perform better.

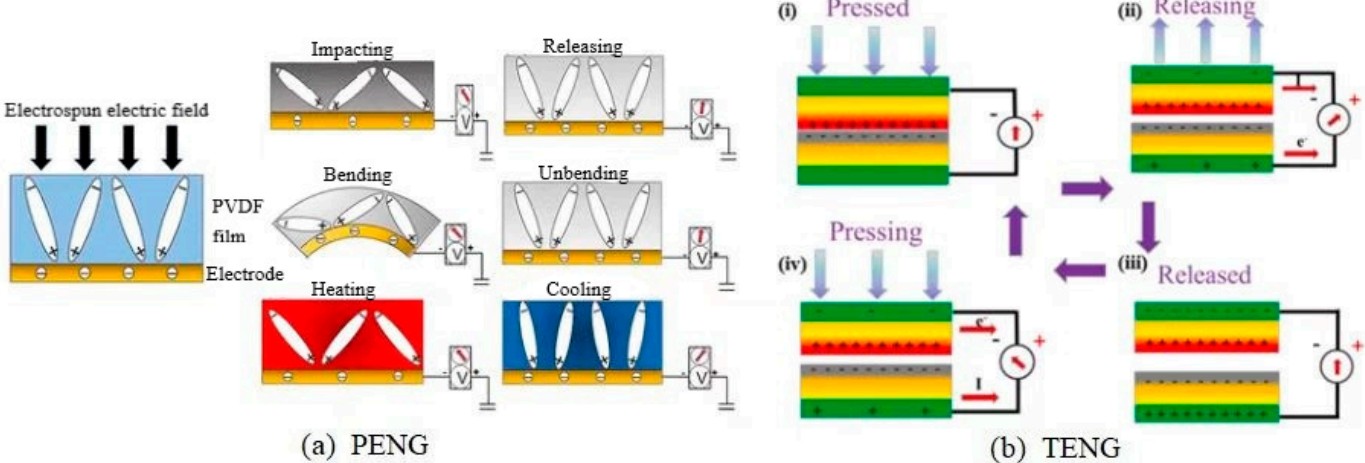

**Figure 2.** (**a**) Working mechanism of PENG. Reproduced from [27] with permission from the American Chemical Society. (**b**) Working mechanism of TENG. Reproduced from [28] with permission from the Elsevier.

### 2.2. TENG

TENG depends on the charge transfer when the two materials contact each other. The working mechanism of TENG is shown in Figure 2b. Such a phenomenon happens widely between two dielectric materials or one dielectric material—one conductor. The dielectric couple can even be the same material pair. After contact with each other, charge generates on the two interfaces, and by following separation, the capacitance change will lead to the induced charge transfer outside the circuit, leading to the current output such as PENG. However, the TENG design is much more flexible. Nowadays, there are mainly four types of TENG. The first is vertical contact-separation mode, in which two materials are in contact and separate vertically, as shown in Figure 3a. In such a mode, contact leads

to charge transfer, and separation causes capacitance change. The second is contact-sliding mode, in which movements parallel with the material surface lead to simultaneous charge transfer and capacitance change, as shown in Figure 3b. The third one is single-electrode mode, similar to the first mode, but only one electrode connected to the ground wire is used, as shown in Figure 3c. Such a mode offers more flexibility. The last one is the freestanding mode, in which a third material moves between two materials, as shown in Figure 3d. The main factor that affects this is electronegativity, as shown in Figure 4. When the electronegativity between the two materials is larger, the output usually becomes larger [29–33].

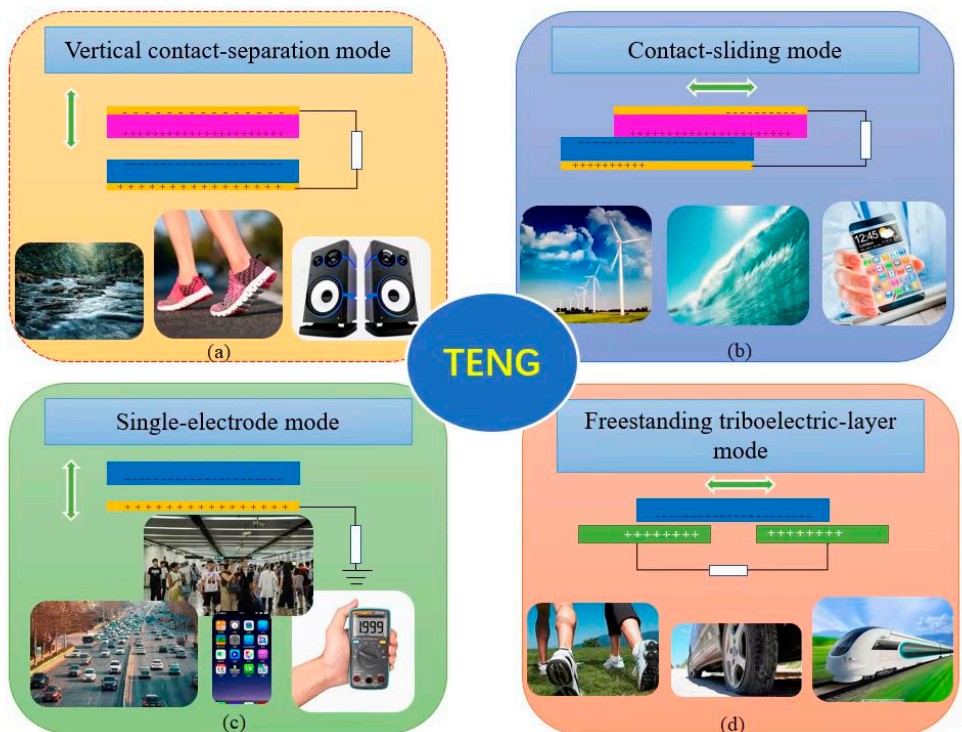

**Figure 3.** Typical modes of TENGs. (**a**) vertical contact-separation mode; (**b**) contact-sliding mode; (**c**) single electrode mode; and (**d**) freestanding triboelectric-layer mode. Adapted with permission from [34]. Copyright 2014 Royal Society of Chemistry.

*2.3. PyNG*

PyNG is different from the commonly observed thermoelectric phenomenon. Such ones work on the pyroelectric phenomenon. If there is a temperature change, the polarization of a piezoelectric material will change, resulting in an induced current between the electrodes. In this case, the typical materials are usually piezoelectric. A combination of energy harvesting techniques can be designed. For inorganic piezoelectric materials, PyNG uses the phase transition temperature (Tc) of ferroelectric materials, and the temperature range should be near it to get the best performance. Phase transition is a common phenomenon in condensed matter. Below such a temperature, ferroelectric order will be maintained, and above such a temperature, no net polarization will be maintained; therefore, the order to disorder transition will leave out the screen charge, and vice versa. In such cases, doping manipulation of Tc or size dependence is commonly used. Such ways will turn Tc to room temperature for the benefit of using such phenomena. For organic materials, Tc will be in a wide temperature range, and in such cases, the pyroelectric output originates from the temperature-induced polarization change and will be lower, but no manipulation of Tc is required [35].

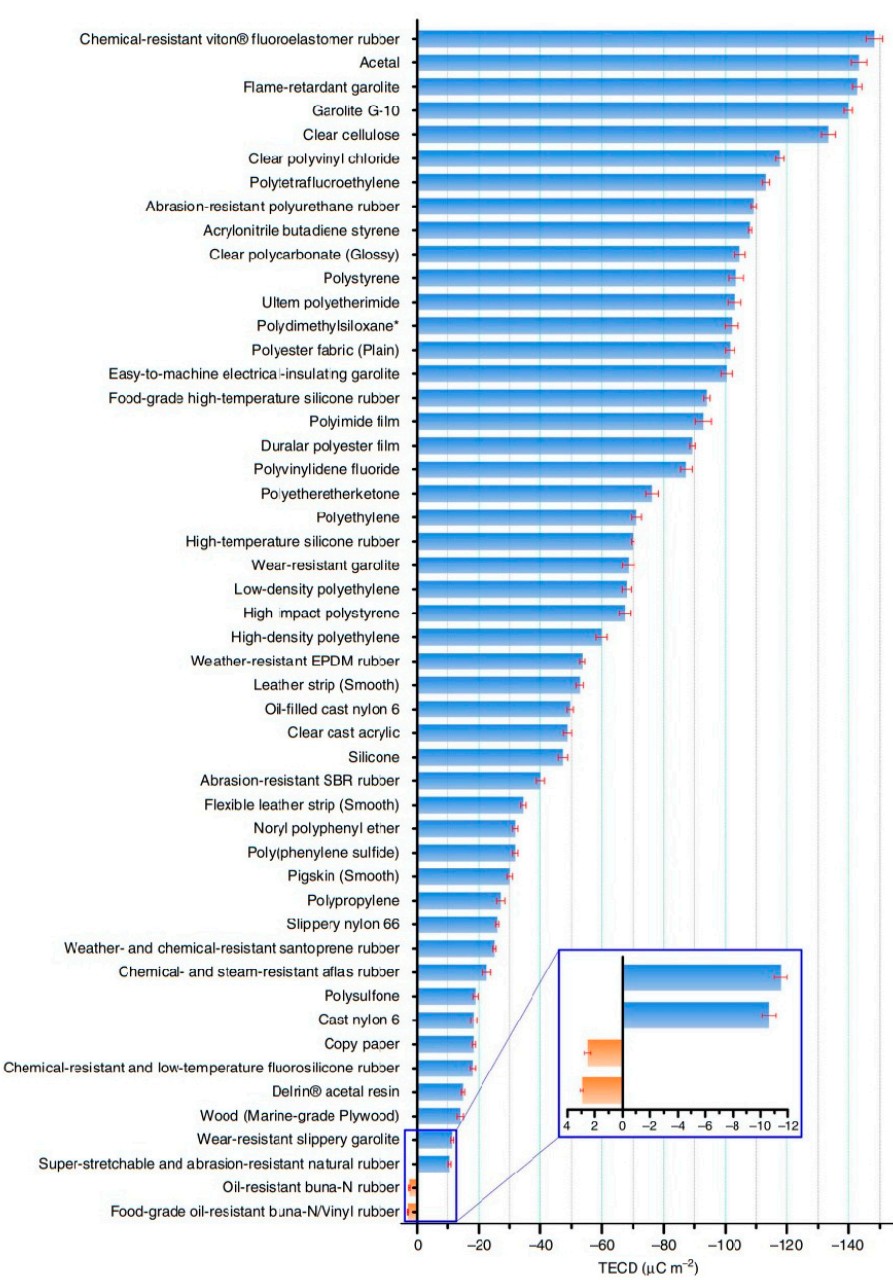

**Figure 4.** Electronegativity of common materials in series. "*" Indicates the average of its molecular weight. Reproduced from [30] with permission from the Nature Portfolio.

## 3. The Methods of Nanogenerators from Waste Plastics

Based on these principles, NGs can be designed. Such design includes material preparation, electrode design, and circuit design for applications.

### 3.1. Material Choice

For material choice, PENGs and PyNGs are usually piezoelectric materials. Such materials include organic and inorganic ones. Usually, inorganic piezoelectric materials have better performance because of their large $d_{33}$ and polarization, which originate from one-century development, and the representative inorganic piezoelectric materials are $BaTiO_3$, PZT, and potassium-sodium niobate (KNN) [36–43]. Here, $BaTiO_3$ is commonly used to study the piezoelectric regulations due to its well-controllable preparation conditions, including single crystal growth. PZTs possess the best piezoelectric output performance and are widely used in daily life, but they cause environmental problems due to the presence of

Pb. In fact, various marvelous piezoelectric materials, including the well-known PMN-PT, are Pb-based ones, so searching for environmentally friendly ones led to the well-designed KNN, a good representation of lead-free piezoelectric materials. For plastic recycling, such inorganic grains can be integrated into the plastics to integrate or enhance the piezoelectric output. For organic ones, the representatives are PVDF and poly-lactic acid (PLLA). PVDF is widely used in daily life, and PLLA is an environmentally friendly plastic. As a representative, PVDF is widely used as a lithium-ion battery separator, vibration sensors, and energy harvesters. It can be solved by acetone and N, N-dimethylformamide (DMF) for further design. The solvents can be spun-coated, knifed, or electrospun into films for designing into applicable forms. Among these, electrospun polarizes PVDF and the film can be directly used, while the other forms of films should be polarized with a high enough electric field with cooling above 100 °C. In fact, some block copolymers can be more easily polarized, including PVDF-TrFE [44–48].

The material choice for TENG is much wider, and the most commonly used pair is nylon to PVDF [49–54]. Here, nylon is widely used, utilizing its strength properties, and it can be solved by formic and acetic acids. The preparation method is similar to piezoelectric ones. By combining such a nylon film with the above-mentioned PVDF film (no polarization is required) with a back electrode, TENG is fabricated in a simple way. In recent years, materials for TENGs have expanded, and the most important direction is liquid-solid TENG, which changes one side into liquids such as water. In fact, the water here can also be waste water, and polytetrafluoroethylene (PTFE) here works as another side. Usually, charge accumulation is required for better performance.

### 3.2. Electrodes

After getting the functional material or material couple, electrodes are the next issue. For a wired one, electrodes should be fabricated aside from the materials to conduct the energy for use. In this scenario, physical vapor deposition (PVD) usually has the best performance. Such methods usually use heat or ions to vaporize the metal and deposit it onto the functional materials. The methods use less metal, and the interface between functional material and metal is tight, so the performance of the device will be good. A compromise method is coating metal-included glues, such as full margin. Such a method requires no expensive PVD instruments but usually costs more metal. In recent years, new electrode materials, including conductive polymers, have also been reported to be applicable for TENG. Apart from the double-electrode design, some electrodeless designs are attracting more and more attention. These methods dig deeper into the mechanism of NGs, and displacement current is well designed. In such a case, the charge is directly used without an outer circuit, making it applicable for simplified applications such as recycled plastics. In the case of piezocatalysis, ultrasonics are used to supply the mechanical energy. The piezoelectric materials induce electric fields with a high enough voltage to lead to catalysis. In the case of a triboelectric design, a smart design is as follows: Aqueous droplets were used to roll down the PTFE. The charge transfer leads to high voltage, and pollutants are degraded. Meanwhile, our recent investigation showed triboelectricity can also happen under water, making it applicable for electrodeless designs such as bare nylon and PVDF contact. In such a case, the output voltage will be much higher.

### 3.3. Ways to Improve the Properties of Nanogenerators

Apart from the basic design of NGs, there are still ways to improve their properties, especially for recycled plastics. The first is functional group design. Polymers are macromolecules integrated by monomers, and the monomers can be changed during polymerization. For example, PVDF-TrFE decreases the electric field required for polarization, and methoxy poly(ethylene glycol) (mPEG) copolymerization with polycaprolactone (PCL) modifies the electronegativity of PCL, leading to larger output performance when it is in contact with nylon [55,56]. The second method is surface modification, including solution treatment, plasma treatment, ion treatment, and so on. For example, the micro-particles

are integrated with the substrate by two-dimensional colloidal self-assembly using the solvent evaporation method. [57,58] Two low-cost materials, including silicon carbide (SiC) and polystyrene (PS), are utilized for modifying the surfaces. To apply the method, nylon (Polyamide 6–6) and fluorinated ethylene propylene (FEP) Teflon are selected, as nylon and FEP polymers show very high positive and negative electron affinities, respectively [59]. The effect of different modified surfaces on significantly enhancing the open-circuit voltage (Voc) and short-circuit current (Isc) of TENG is studied. The proposed method can be applied for large-area fabrication of micro-structures on various polymeric films, which can also improve the hydrophobicity of the surfaces. The good durability of the modified TENG performance is also proved by applying numerous cyclic loads. In addition, the practical application of modified TENG for energy harvesting applications is shown by charging different storage units. In another work, protons accelerated to 150 KeV were used to treat polyimide (PI). They penetrate the film and change the surface properties of the 2 μm area, leading to an increase in output when in contact with liquid metal [60]. The third method is surface contact modification. For example, an electrospun membrane can be designed by a roller collector, leading to the alignment of the fibers. Then the micro contact area will be different using two membranes with an ordered fiber arrangement in case the relative angle changes. The new design method is still emerging, but in cases where recycled plastic requires less treatment, the surface modification method will be the best choice [61,62].

## 4. Electrocatalytic Methods of Nanogenerators from Waste Plastics

Traditional methods of electrochemistry have been used for the removal of contaminants from wastewater, which involve electrochemical reduction, electrocatalytic oxidation, electrodialysis, and electroflotation. Electrocatalysis is commonly used for the treatment of biorefractory organic wastewater, as it is considered to be one of the most widely used electrochemical advanced oxidation processes (EAOPs). Its high effectiveness and reliability for treating wastewater have attracted considerable attention in recent times. By utilizing the electric energy obtained from NGs to drive electrocatalytic oxidation, it is possible to substantially reduce the need for external energy input and enhance the applicability of NGs for treating wastewater on a larger scale. Numerous studies have confirmed the effectiveness of using NGs to eliminate organic pollutants from wastewater while avoiding the need for a conventional electrochemical configuration. Therefore, considering the benefits of energy conservation and environmental preservation, the combination of NGs and electrocatalytic technology represents a highly advantageous approach to treating wastewater [63].

PENGs were the first reported degradation of pollutants. This method uses PENGs to convert mechanical vibration into electricity to degrade the pollutant. In fact, such an area overlaps with piezocatalysis. Compared with other methods, catalysis with PENGs has the advantage of easy manipulation. For example, ions can be easily doped to improve the conductivity of catalysis, and hetero junction can also be designed for the separation of carriers. Yu Yang fabricated a Ca-doped $BaTiO_3$ membrane, and such a membrane was used to degrade Congo Red. To identify catalysis from adsorption, tert-butanol (TBA, ●OH scavenging agent) and ethylenediamine diabetic acid tetrasodium dihydrate sodium salt (EDTA-2Na, h scavenging agent) were used, and if the change of color stops, it can be determined to be the degradation of dye [64]. The degradation and catalytic performance recovery test results of Congo Red are shown in Figure 5.

Paired with the TENG, the mechanical energy generated during wastewater treatment processes can be transformed into renewable electrical energy, which is then used to power the electrocatalysis process. With this approach, electrocatalysis becomes a self-sustaining system that enhances the efficiency of catalytic degradation in wastewater treatment. Previous research has indicated that self-powered electrochemical systems utilizing TENG technology have been successfully implemented in various applications, including water splitting, protecting against water pollution, providing cathodic protection, facilitating

seawater desalination, and enabling electrochromic reactions. The low production costs, ease of fabrication, and outstanding electrocatalytic performance of TENG technology render it an attractive option for combined use with electrocatalysis in wastewater treatment processes. For example, Ye Chen designed a self-powered EF system based on the advantages of electric Fenton (EF) and the flexible design of triboelectric nanogenerators (TENG) and biomass carbon materials. It is driven by a sturdy and flexible multi-layer TENG (RFM TENG), using carbon materials derived from magnolia flowers as the cathode for oxygen reduction [65–67].

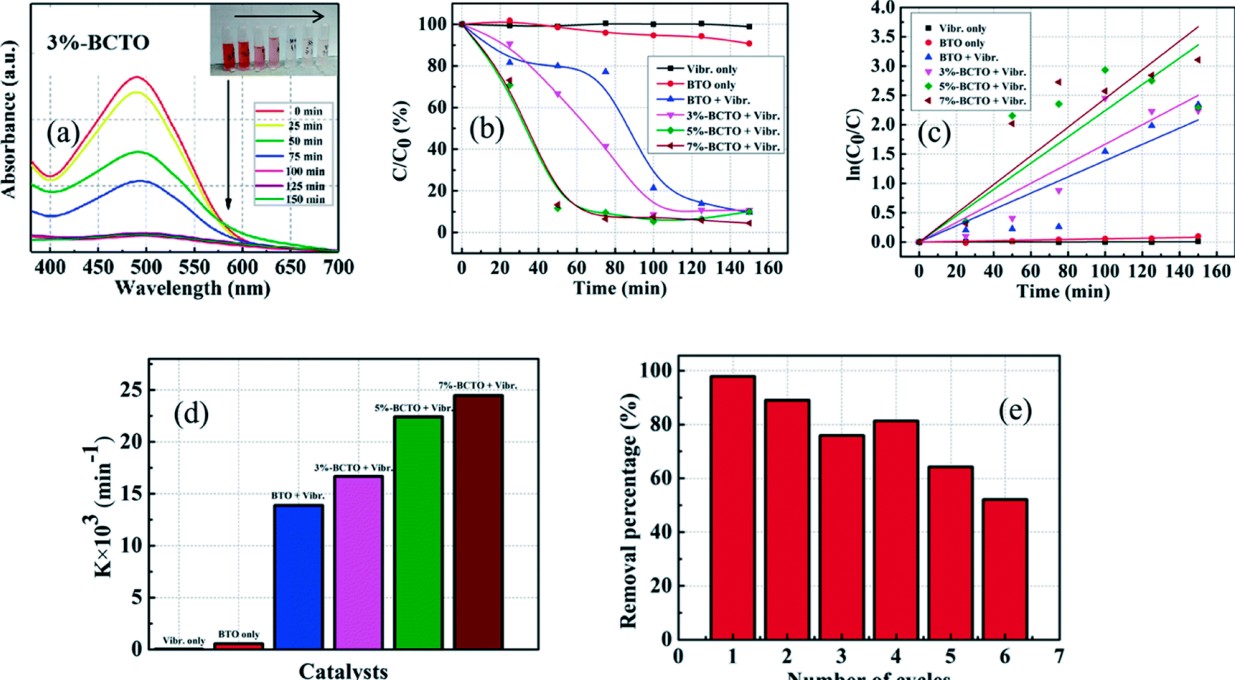

**Figure 5.** Piezocatalysis performance of Ca doped $BaTiO_3$. (**a**) Spectrum recoding of degrading dye. The left to right samples were listed according to the catalytic time, from earlier to later. (**b**) Time dependence and (**c**) kinetic analysis of it catalyze performance. (**d**) Comparison results. (**e**) Cycle results. Reproduced from [64] with permission from the Royal Society of Chemistry.

In the photo-catalytic system, there is a rapid recombination of charge carriers generated by light, which can decrease the quantum yield responsible for producing reactive oxygen species [63,68,69]. When compared to photo-catalysis, photoelectric catalysis presents a greater challenge for the recombination of photogenerated electrons and holes, resulting in increased effectiveness. Introducing an electric field to the process of photoelectric catalysis can enhance the redox potential available for the degradation of pollutants. TENGs produce electricity that meets the prerequisites for high voltage and low current, which can aid in the catalytic process. Therefore, the close integration of TENGs and photoelectric catalysis is capable of achieving high removal rates for wastewater treatment without any energy input. Photoelectric catalysis can be categorized into two distinct types: electroassisted photocatalysis and photoassisted electrocatalysis, such as photoelectric-Fenton. The photocatalyst is fixed to an electrode in the form of a photoanode. When the photoanode is exposed to sunlight, electrons are stimulated to move from the valence band to the conduction band, while holes are produced in the valence band. The use of an external electric field can result in greater efficiency in electron-hole separation. Utilizing TENG-based photocatalysis offers a practical and simple method for the efficient elimination of persistent organic pollutants during the treatment of wastewater. During electroassisted photocatalysis, the presence of $TiO_2$ and other materials can hasten the recombination process of photogenerated electrons, thereby lessening the quantum yield related to the creation of reactive oxygen species

(ROS). In electroassisted photocatalysis, the photocatalyst is affixed to a conductive support and functions as a photoanode in a two-electrode battery configuration. Under sunlight, when the photoanode is illuminated and the energy of the incident photon exceeds the energy bandgap of the semiconductor photocatalyst, electrons are excited from the valence band -VB to the conduction band -CB, leading to the generation of holes in the -VB. The oxidizable organic matter is oxidized to inorganic matter by the transfer of electrons to $h_{VB}^+$. As a result, a conjugation reaction takes place between $H_2O$ and active electrons, and between OH– or $O_2$ and $h_{VB}^+$ to generate ●OH and other ROS. These ●OH then mineralize various waste plastics, effectively removing them from the environment. By utilizing TENG technology, the introduction of an external electrical bias can enhance the separation of charges and subsequently enhance the overall efficiency. TENGs also benefit from their wide material choice, and in such a case, polluted water can also be one friction side. In such a case, the charge transfer will lead to an improvement in carrier change, leading to enhanced degradation of dyes. For example, Song et al. used PTFE, which is a well-known material for designing liquid-solid NGs, and crystal violet was just used to simulate common dye wastes, and such a simple water droplet tumble showed very good catalysis performance, as shown in Figure 6 [70]. By using wasted plastic TENGs, photocatalyst can be loaded, achieving better performance. For example, Su et al. conducted a study that utilized a Teflon-Al-based triboelectric nanogenerator (TENG) to degrade methyl orange (MO), which is a primary component of waste plastics, using $TiO_2$ under ultraviolet irradiation. The photo-generated electrons and holes migrate to the surface of $TiO_2$ particles and act as redox sources, effectively reacting with adsorbed reactants to produce superoxide radical anions, hydrogen peroxide, and hydroxyl radicals—all of which are involved in the oxidation of waste plastics. The TENG generates electricity output that enhances the photodegradation of MO when used in tandem with $TiO_2$ nanoparticles. This is because the electric field generated by the TENG helps to separate and suppress the recombination of photo-generated electrons and holes. Due to the photoelectrical coupling, the degradation percentages of MO after 120 min with and without TENG assistance were 76% and 27%, respectively. These results indicate that the fabricated TENGs have the potential for use in the degradation of waste plastics [71].

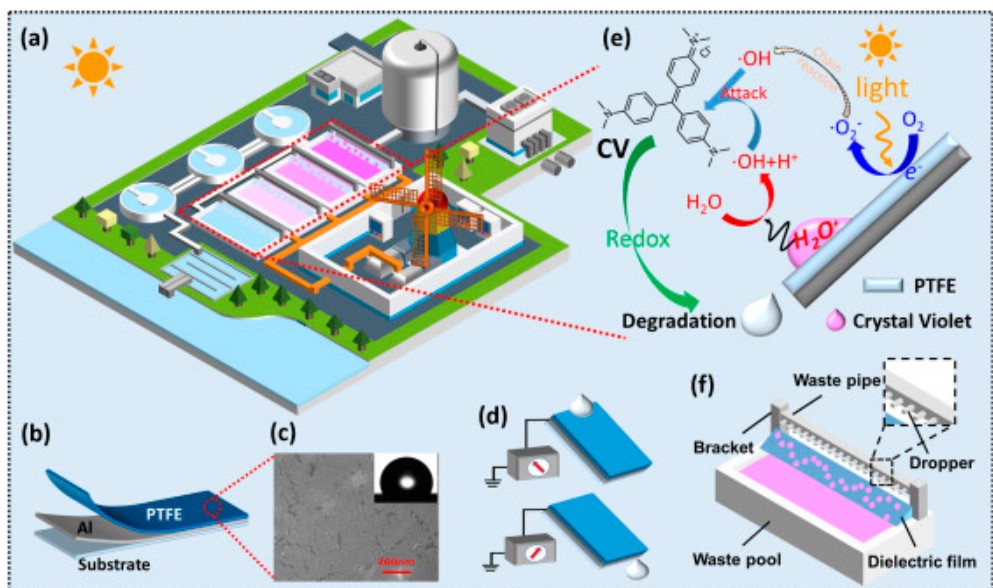

**Figure 6.** Tribocatalysis by liquid-solid contact. (**a**) Application demonstration, (**b**) solid side structure, (**c**) SEM image of solid side, (**d**) working principle, (**e**) reaction process, and (**f**) experiment set up. Reproduced from [70] with permission from the Pergamon-Elsevier Science Ltd.

Scholars around the world have been focusing on the integration of photocatalysis and biodegradation, also known as ICPB, in recent years. This innovative technology

combines the strengths of both biotechnology and photocatalysis, offering a low-cost, environmentally friendly, and sustainable approach for the treatment of pollutants [72]. ICPB involves the use of photocatalytic materials, porous carriers, and biofilms. The porous carrier's surface allows for the conversion of refractory pollutants into biodegradable pollutants via photocatalysis. Meanwhile, the microorganisms on the biofilm inside the carrier facilitate the degradation of biodegradable pollutants into $CO_2$ and $H_2O$ [73].

Through photoelectric catalysis outside the biofilm carrier, refractory pollutants can be broken down into biodegradable pollutants, thereby increasing the BOD/COD ratio [74]. Subsequently, microorganisms inside the carrier can degrade the biodegradable pollutants into inorganic matter. The TENG-IPECB system relies on the mechanism of intimate coupling between self-powered photoelectric catalysis and biodegradation. In this system, waste gas produced during the treatment of pollutants causes the internal materials of the TENG device to undergo friction, contact, and separation. A biological community is established in the porous carrier, which forms a biofilm used to degrade pollutants. The outer surface of the carrier is coated with photoelectric materials, while the inner surface is colonized by microorganisms. When the photoelectric materials are exposed to sunlight, holes ($h^+$) are generated and electrons ($e^-$) are formed. The electric energy generated by the TENG is then applied to IPECB cells. This electric energy is used to power the photoelectric catalysis system, creating a large potential difference between the photoanode and cathode. The flow of electrons from the photoanode to the cathode, under this potential difference, makes it difficult for $e^-/h^+$ pairs to recombine. Hence, refractory pollutants are oxidized by the $h^+$ into biodegradable pollutants [75].

Microorganisms attached to the biofilm inside the porous carrier play a crucial role in degrading biodegradable pollutants into inorganic products. By significantly improving the efficiency of photocatalysis, the TENG-IPECB system can reduce the burden on subsequent biodegradation, leading to improved removal efficiency of contaminants throughout the process. In their study, Yasmina et al. utilized $TiO_2/Ti$ composite electrodes, along with biofilms, to degrade phenol via photocatalysis [76]. The results showed that the microorganisms attached to the $TiO_2$ electrode acted as a biofilm and that the transfer of electrons from the electrode to the microorganisms facilitated the formation of the biofilm on the electrode. Ultimately, phenol was oxidized into absorbable organic matter on the biofilm, serving as the carbon source for the growth of microorganisms. In addition, through magnetron co-sputtering, Professor Kuru found that doping Mg with UV light can improve the photocatalytic activity of ZnO thin films. It was found that the photocatalytic efficiency of ZnO and MgZnO thin films after heat treatment is the highest at 400 degrees Celsius [77]. Moreover, the thickness of MgZnO film has an effect on its structure and photocatalytic performance [78]. Table 1 summarizes the content of the article, including the types of nanogenerators, the materials used to manufacture NG, the types of electrodes, and the advantages of their applications.

**Table 1.** Summarizes the content of the article, including the types of nanogenerators, the materials used to manufacture NG, the types of electrodes and the advantages of their applications.

| Type | Materials | Electrode | Advantage | Ref. |
|---|---|---|---|---|
| TENG | PTFE, PVDF | Magnolia carbon material for the cathode or some metal material | Low cost, simple preparation method, and high efficiency electrocatalysis. The electric energy generated by the teng meets the requirements of high voltage and low current, which can assist the catalytic process. | [66,67,70] |
| PENG | $BaTiO_3$ film doped with Ca, PZT, TVDF, $BaTiO_3$, KNN | with the cathode carbon materials from TM and EDTA-2Na | The adsorption catalysis was evaluated, and the polymer catalysis had the advantage of being easy to operate. | [36–43,65] |
| PyNG | PZT, TVDF, $BaTiO_3$, KNN | Ferroelectric material | It converts heat into electrical signals. | [36–43] |

## 5. Conclusions

### 5.1. Conclusions

In short, catalysis using NGs is an emerging area. Various catalysis methods, including piezocatalysis, tribocatalysis, and pyrocatalysis, are the three main methods. They can be used directly to collect mechanical energy or to assist other catalysis. Despite intensive research efforts, the frontier of today still has a long way to go before real applications. The main challenge is the increased degradation of multiple types of pollutants. The recycled plastics decrease the second pollutant, but obviously, better design for the degradation of more pollutants increases the application value. To achieve this goal, mechanism research is fundamental, and better design based on the given mechanism is the key factor for area improvement. At the same time, the degradation of micro-plastics or even the NG itself is a further challenge. Despite the catalysis improvement, the NG design based on recycled plastics is also attractive. Some ways, such as electrostatic selection of plastics, may help collect plastics with better triboelectric performance. Meanwhile, electret is also a novel NG type, and plastics are novel candidates to be charged into electrets, and the best one is PI. Designing applications based on emerging NG technologies such as this will result in better performance. To conclude, NGs are based on common plastics, which is very easy to achieve with recycled plastics. NGs can be designed into various catalyzers applicable in electrocatalysis, degrading the pollutants and solving the water pollution problem.

### 5.2. Expectations for the Future

Firstly, there will be a classification method emerging in the near future for the systematic fabrication of tribo functional layers. For example, electrostatic classifications may help organize recycled plastics into positive and negative sides, and the organized parts may work even without further purification. Secondly, efficient surface modification methods will emerge, as will various optimization methods, such as increasing electronegativity/electropositivity or saving costs while maintaining the best performance. What's more, surface roughness modification is essential considering the huge effect of surface roughness on triboelectrification.

**Author Contributions:** Conceptualization, Q.Z. and Q.P.; validation, X.W., W.L. and Y.L.; writing—original draft preparation, Q.Z. and Q.P.; writing—review and editing, X.W., W.L. and Y.L.; project administration, X.W.; funding acquisition, Q.Z. All authors have read and agreed to the published version of the manuscript.

**Funding:** This research was supported by the Shandong Provincial Natural Science Foundation, China (grant number ZR2019PEE011).

**Data Availability Statement:** Date availability is not applicable to this article as no new data were created or analyzed in this study.

**Conflicts of Interest:** The authors declare no conflict of interest.

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
