# Peer review of "Waste Plastic Recycling Upgrade Design Nanogenerator for Catalytic Degradation of Pollutants"

_catalysts, doi:10.3390/catal13061019_

Round 1

Reviewer 1 Report

1. There are generally 5 keywords.

2. The conclusion section should supplement expectations for the future of this field.

3. It should be introduced that Photocatalytic Methods of Nanogenerators from Waste Plastics.

4. Some pictures are not very clear, for example, Figure 4.

5. Suggest adding some discussion on the mechanism section about Catalytic Degradation of Pollutants.

6. The reference format needs to be unified.

it is ok.

Author Response

We appreciate the professional judgment on our work and have carefully revised the manuscript based on the comments.

Reviewer 2 Report

The authors have written a fairly comprehensive review of nano generators technology for catalytic degradation of pollutants for plastic recycling.

The structure of the manuscript is sound and well organized according to mechanisms of nano generators basically narrowed down and focused on piezoelectric, triboelectric, and pyroelectric and their classifications according to electrode type and application. The structure and focus is the paper makes it suitable as a review article. It is recommended that this paper be published in Catalysts after the following issues have been addressed by the authors:

1. There are nano generators used for photodegredation using solar UV light that are based on ZnO and its nano composite MgZnO. The authors should consider adding some of those to section 4 of the paper.

2. To make the review article more manageable for the readers, the authors should include a table at the end of section 4 that summarizes the contents of the review article with the following suggested columns: column 1: nano generator type (TENG, PENGUIN, PyENG); column 2: Material Used to Fabricate NG; column 3: Electrode Type; column 4: Application and Advantages; column 5: Reference Number.

3. Minor modifications on the style and usage of English is needed all throughout the manuscript. See comments on English usage.

Minor modifications on the style and usage of English is needed all throughout the manuscript. See comments on English usage. Examples are: page 1, Introduction first paragraph: “It‘s attracts wide attentions for its simple preparation process, …” should be “It attracts…”, another example is the word “principal” should be replaced with “principle”. Another example is the agreement of subject-verb number like “electrodes is” should be “electrodes are.” There are plenty of these errors abound the manuscript and the authors are advised to go through the entire manuscript to correct them.

Author Response

We appreciate the professional judgment on our work and have carefully revised the manuscript based on the comments. Please see the attachment.

Round 2

Reviewer 2 Report

The authors have addressed all the issues addressed by the reviewer. It is recommended that the review article be published is Catalyst.